# Aluminium in Brain Tissue in Epilepsy: A Case Report from Camelford

**DOI:** 10.3390/ijerph16122129

**Published:** 2019-06-16

**Authors:** Matthew Mold, Jason Cottle, Christopher Exley

**Affiliations:** 1The Birchall Centre, Lennard-Jones Laboratories, Keele University, Staffordshire ST5 5BG, UK; m.j.mold@keele.ac.uk; 2School of Medicine, David Weatherly Building, Keele University, Staffordshire ST5 5BG, UK; jasoncottlee@gmail.com

**Keywords:** aluminium in brain tissue, epilepsy, aluminium-specific fluorescence, occipital calcifications, tau pathologies, Camelford in Cornwall

## Abstract

(1) Introduction: Human exposure to aluminium is a burgeoning problem. In 1988, the population of the Cornish town of Camelford was exposed to exceedingly high levels of aluminium in their potable water supply. Herein we provide evidence that aluminium played a role in the death of a Camelford resident following development of late-onset epilepsy. (2) Case summary: We have measured the aluminium content of brain tissue in this individual and demonstrated significant accumulations of aluminium in the hippocampus (4.35 (2.80) µg/g dry wt.) and the occipital lobe (2.22 (2.23) µg/g dry wt., mean, SD, *n* = 5), the latter being associated with abnormal calcifications. Aluminium-specific fluorescence microscopy confirmed the presence of aluminium in both of these tissues and made the consistent observation of aluminium-loaded glial cells in close proximity to aluminium-rich cell/neuronal debris. These observations support an inflammatory component in this case of late-onset epilepsy. Congo red failed to identify any amyloid deposits in any tissue while thioflavin S showed extensive extracellular and intracellular tau pathologies. (3) Discussion: We present the first data showing aluminium in brain tissue in epilepsy and suggest, in light of complementary evidence from scientific literature, the first evidence that aluminium played a role in the advent of this case of late-onset adult epilepsy.

## 1. Introduction

In 1988, twenty tonnes of aluminium sulphate, destined for a holding tank at Lowermoor Water Treatment Works, was inadvertently added directly into the potable water supply serving the town of Camelford, Cornwall in the United Kingdom. This event is notoriously known as the Camelford water-poisoning incident and up to 20,000 people were exposed to very high levels of aluminium in their drinking water both immediately and for several weeks thereafter. While there have been a number of government inquiries into this incident, the final one having reported in 2013, none of the subsequent recommendations for further research has been implemented. There have been two published coroner-led investigations of putative victims of Camelford and in both cases unusual neuropathologies were associated with coincidentally high levels of aluminium in brain tissue [1,2,3,4]. Herein we present findings relating to a third coroner’s investigation of someone who was exposed to the Camelford incident and died of asphyxiation associated with an epileptic fit. Quantitative data pertaining to the aluminium content of brain tissue were included in the coroner’s report in 2014. However, qualitative imaging of aluminium in brain tissue was not available at that time, as a method using aluminium-specific fluorescence has only recently been developed [2]. This new method has now been validated and has been used to unequivocally identify aluminium in brain tissue involving sporadic Alzheimer’s disease [2], familial Alzheimer’s disease [5], autism [6], multiple sclerosis [7] and cerebral amyloid angiopathy [4]. We have used this method herein to complement the quantitative measurements and now present the first data and images of aluminium in brain tissue involving epilepsy.

## 2. Case Summary

A 60-year-old man died of asphyxiation brought on by an epileptic fit. The deceased was described by the coroner as a victim of the Lowermoor Treatment Works, Camelford. Subsequent to his exposure to aluminium in 1988 he was treated for psychotic illnesses and 5 years prior to his death he developed epilepsy that proved difficult to manage therapeutically. Frozen and fixed brain tissues were provided by University Hospitals Plymouth, NHS Trust, UK and sent to Keele University at the coroner’s request to investigate their content and distribution of aluminium. Written informed consent was obtained from the patient‘s next of kin for publication of this case report and any accompanying images. A copy of the written consent is available for review by the Editor-in-Chief of this journal.

Frozen brain tissue from frontal, occipital, parietal and temporal lobes and hippocampal tissue were processed for aluminium determination according to well-established protocol [8] described only briefly here. Thawed tissue was cut using a stainless-steel blade into pieces of approximately 0.5 g wet weight. Weighed tissues were dried in an incubator at 37 °C to a constant dry weight before being digested in a microwave (MARS Xpress CEM Microwave Technology Ltd., (Buckingham, UK) in a mixture of 1 mL 15.8 M HNO_3_ (Fisher Analytical Grade) and 1 mL 30% *w/v* H_2_O_2_ (BDH Aristar Grade). Digests were then made up to 5 mL with ultrapure water (cond. < 0.067 μS/cm). The total aluminium in digests was measured by TH GFAAS using matrix-matched standards. Data are expressed herein as μg Al/g tissue dry weight, following subtraction of a method blank [8].

Paraffin-embedded human brain tissue blocks mounted on Tissue-Tek^®^ Uni-cassettes^®^ (Sakura Finetek Europe B.V., Alphen ann den Rijn, Netherlands) were provided for frontal, parietal, temporal and occipital lobes and hippocampal tissue. Tissue blocks were cooled on wet ice for 10 min and sectioned using a Leica RM2025 rotary microtome equipped with Surgipath^®^ DB80 LX low-profile blades (Leica Microsystems, Newcastle Upon Tyne, UK). Serial sections were prepared as ribbons at a thickness of 5 μm and floated onto ultrapure water (cond. <0.067μS/cm) at 40 °C. Serial sections were allowed to flatten out for 30 s, prior to floating onto numbered glass SuperFrost^®^ Plus adhesions slides (Thermo Scientific, Loughborough, UK). Mounted sections were stored vertically at ambient temperature overnight and heated immediately before use at 62 °C for 20 min to remove excess paraffin. Slides were allowed to cool to ambient temperature, prior to conducting deparaffinisation procedures. All chemicals, unless otherwise stated, were from Sigma Aldrich, Gillingham, UK. 

Lumogallion (4-chloro-3-(2,4-dihydroxyphenylazo)-2-hydroxybenzene-1-sulphonic acid, TCI Europe N.V., Zwijndrecht, Belgium) staining was performed as previously described [2,4,6]. Briefly, sections were de-waxed via transfer into fresh Histo-Clear (National Diagnostics, Atlanta, GA, USA) and rehydrated through an ethanol (HPLC grade) gradient from 100 to 30% *v/v*, prior to rehydration in ultrapure water (cond. <0.067 μS/cm). Staining was performed in moisture chambers in which rehydrated sections were outlined with a hydrophobic PAP pen to which 200 μL of 1 mM lumogallion in 50 mM PIPES, pH 7.40, was added or buffer only for non-stained autofluorescence sections. Sections were incubated at ambient temperature for 45 min. away from light, prior to rinsing in PIPES buffer and mounting with Fluoromount™. 

Congo red staining was performed via immersion of rehydrated sections into 0.5% *w/v* Congo red in 50% *v/v* ethanol, for 5 min. Sections were differentiated in a solution of 0.2% *w/v* potassium hydroxide in 80% *v/v* ethanol for 3 s and rinsed in ultrapure water for 30 s. Sections prepared in this way were mounted with Faramount (Agilent Dako, Stockport, UK).

Dewaxed and rehydrated brain tissue sections for thioflavin S (ThS) staining were outlined with a hydrophobic PAP pen and 200 μL of *ca* 0.075% *w/v* ThS in 50% *v/v* (HPLC) grade ethanol was added. Staining was performed in humidity chambers away from light at ambient temperature for 8 min. Stained sections were subsequently washed twice for 10s in fresh solutions of 80% *v/v* (HPLC) grade ethanol, prior to rinsing in ultrapure water for 30 s. All sections were mounted with glass coverslips using the aqueous mounting medium Fluoromount™. Sections were allowed to cure overnight at 4 °C away from light, prior to their analysis via fluorescence microscopy. 

All microscopy was performed through use of an Olympus BX50 fluorescence microscope, equipped with a vertical illuminator and BX-FLA reflected light fluorescence attachment (mercury source). Lumogallion fluorescence, characteristically bright orange to intense yellow depending upon the concentration of available Al^3+^, and respective autofluorescence in the absence of the fluorophore was visualised using a U-MNIB3 (excitation filter: 470–495 nm, dichromatic mirror: 505 nm, longpass emission filter: 510 nm) fluorescence filter cube (Olympus, UK). Light exposure and transmission settings were fixed across respective staining conditions. Brain tissue sections were sequentially scanned for the identification of lumogallion-reactive aluminium and where positive fluorescence was identified complementary autofluorescence was assessed on a non-stained serial section. Apple green birefringence of Congo red stained amyloid deposits was obtained using a U-POT drop-in polariser and a U-ANT transmitted light analyser (both from Olympus, UK) and assessed through sequential screening. ThS imaging was performed via use of a U-MWBV2 fluorescence filter cube (excitation filter: 400–440 nm, dichromatic mirror: 455 nm, longpass emission filter: 475 nm). All images were acquired using the CellD software suite (Olympus, SiS Imaging Solutions, SiS, GmbH). Bright-field and fluorescence channels were merged using Photoshop (Adobe Systems Inc., San Jose, CA, USA).

The aluminium content of brain tissues (mean, SD, *n* = 5) were 4.35 (2.80) μg/g dry wt. in the hippocampus and 0.78 (0.48), 2.22 (2.23), 0.81 (0.59) and 0.50 (0.35) μg/g dry wt. in the temporal, occipital, frontal and parietal lobes, respectively. The aluminium content was high in the hippocampus and in the occipital lobe two of the five tissue replicates had a high content of aluminium (5.51 and 3.54 μg/g dry wt.); these two tissues were characterised by a highly mineralised/calcified appearance.

Lumogallion staining of the frontal lobe revealed both extracellular cortical deposits of aluminium as well as intracellular punctate deposits within glial cells (Figure 1a). Non-stained adjacent serial sections revealed a dull green autofluorescence emission (see Appendix A). Lumogallion-reactive aluminium was not clearly identified in parietal lobe tissue while lipofuscin, evident as a yellow fluorescent pigment, was identified in both stained (Figure 1b) and non-stained tissues. Aluminium was found in the occipital lobe, both as punctate deposits in microglial cells and associated with cellular debris (Figure 2a). Aluminium was also observed in astrocyte-like cells in the occipital cortex, again as punctate deposits of approximately 1 μm in size (Figure 2b). The identical regions in non-stained adjacent sections revealed weak green fluorescence emission and intraneuronal accumulations of lipofuscin (see Appendix A). Extracellular deposits of aluminium were only observed in white matter regions of the temporal lobe (Figure 3a). Intracellular aluminium was predominantly identified within the temporal cortex, noted via punctate orange fluorescence within inflammatory cells in the vessel wall (Figure 3b). Serial sections for autofluorescence confirmed the absence of positive staining attributed to aluminium in the lumogallion-stained tissue sections (see Appendix A). In the parahippocampal gyrus, an intense punctate intracellular aluminium fluorescence was observed in glial cells in close proximity to extracellular deposits of aluminium (Figure 4a). Intracellular aluminium was also shown in glial cells exhibiting astrocyte-like processes in the hippocampus (Figure 4b). Analysis of non-stained sections for autofluorescence confirmed the presence of aluminium in lumogallion-stained sections (see Appendix A). 

Congo red staining of the hippocampus and the frontal, parietal, occipital and temporal lobes revealed the complete absence of Congo red-reactive β-amyloid, under both bright field and polarised light illumination modes. However, in the occipital lobe, spherulite-like structures were identified under polarised light by Maltese cross diffraction patterns emitting a characteristic white birefringence (Figure 5a,c). Under bright field illumination, ovoid bodies of a globular morphology, with an outer diameter of 13.6 ± 2.9 μm (mean ± SD, *n* = 36), were found deposited along the stria of Gennari within the visual cortex of the occipital lobe (Figure 5b,d). The spherulite-like structures were not found co-located with aluminium upon lumogallion staining of adjacent serial tissue sections. 

Thioflavin S (ThS) staining of all lobes of the cerebral cortex and the hippocampus revealed the presence of flame-like neurofibrillary tangles (NFTs), as identified via an intense green fluorescence emission [9] at the periphery of cortical brain regions (Figure 6a–h). While NFTs emanating from neuronal cell bodies were readily identified, their presence in the temporal lobe was sparse in comparison to all other lobes investigated. Ring-like tangles of presumably tau protein were also apparent within pia matter of the frontal (Figure 6a,b) and parietal (Figure 6c,d) cortex and within the hippocampus (Figure 6h). Furthermore, the same ring-like structures were predominantly identified within epithelial cells of the choroid plexus (Figure 6h), occasionally depicting a flame-like morphology as with those identified as NFTs in grey matter regions. No evidence for the direct co-localisation of aluminium with ThS-reactive structures (NFTs) was identified.

## 3. Discussion

We have made the first quantitative analyses of aluminium in brain tissue involving human epilepsy and supported these measurements with images identifying the presence and location of aluminium. The hippocampus was the primary target for the accumulation of aluminium and quantitative data were supported by imaging showing both the presence of extracellular aluminium and aluminium located within glial cells (Figure 4). Glial-like cells loaded with aluminium and close to aluminium-rich cellular/neuronal debris were consistent observations in these epilepsy brain tissues. Congo red failed to identify any diffuse or β-sheet amyloid in these brain tissues. Congo red did identify many spherulite-like structures in the visual cortex of the occipital lobe (Figure 5) an observation that was coincident with the occurrence of heavily mineralised, presumed calcified, tissue in the same region. The latter tissues were also high in aluminium, although the co-localisation of aluminium and spherulites could not be confirmed by lumogallion staining. ThS staining revealed numerous extracellular and intracellular NFT-like deposits throughout these epilepsy tissues (Figure 6), although again, co-localisation of these structures with aluminium was not demonstrated.

While a single case study cannot implicate human exposure to aluminium in the aetiology of epilepsy, there are additional clues within the published scientific literature that together with the data presented herein do make this link. Most prominent amongst the additional clues is the high incidence of epilepsy in autism [10]. In a recent study demonstrating high aluminium in autism brain tissue, several brain donors suffered from epilepsy [6]. Dravet syndrome is a refractory epileptic syndrome that is heavily linked to inflammation following vaccination involving aluminium adjuvants [11]. Aluminium overload in renal-transplant patients has been shown to be a predisposing factor for epileptic seizures [12]. Aluminium in blood has also been shown to be higher in individuals with epilepsy [13] and there is an example where regular treatment with the aluminium chelator DFO concomitantly lowered blood aluminium concentrations and abolished epileptic seizures in an individual with Alzheimer’s disease [14]. Finally, another piece of striking evidence linking aluminium with epilepsy is the use of aluminium salts to induce epilepsy in non-human primate models of the disease [15]. Perhaps herein lies the strongest evidence for a possible role for aluminium in epilepsy. 

The hippocampus is an important target in epilepsy and our observation of a high concentration of aluminium in hippocampal tissue combined with evidence of aluminium-induced inflammation supports a role for aluminium in this case. We also observed another feature of epilepsy: significant tissue calcification in the occipital lobe [16], which herein was coincident with high levels of aluminium. Contrary to some suggestions in the literature for adult onset epilepsy, we did not identify any amyloid deposition [17] in this case while NFT-like structures both extra- and intracellular were prevalent throughout all four lobes and the hippocampus [18].

## 4. Conclusions

We have presented the first data on aluminium content of brain tissue in epilepsy and the first direct evidence that aluminium may play a role in this disease. This also represents the third coroner-led investigation into the role played by aluminium in victims of the Camelford water-poisoning incident. This is the third example in which aluminium is suggested to have played a role in an exposed individual’s death and should now be a catalyst for re-opening investigations into this notorious incident.

## Figures and Tables

**Figure 1 ijerph-16-02129-f001:**
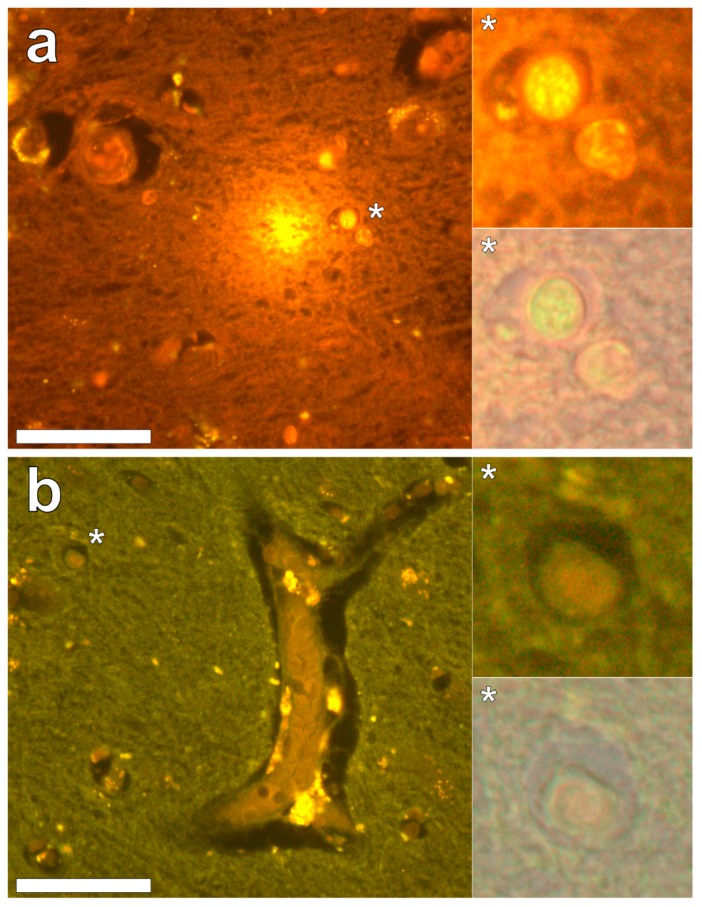
Aluminium-loaded cells in the frontal cortex, morphologically compatible with glia, identified by punctate orange fluorescence, are in close proximity to aluminium-rich extracellular debris (**a**) Lumogallion staining of the parietal lobe (**b**) is negative for the presence of aluminium with cortical lipofuscin deposits highlighted via yellow fluorescence. Magnified inserts are denoted by asterisks, including a bright-field overlay in the lower panels. Magnification × 400, scale bars: 50 μm.

**Figure 2 ijerph-16-02129-f002:**
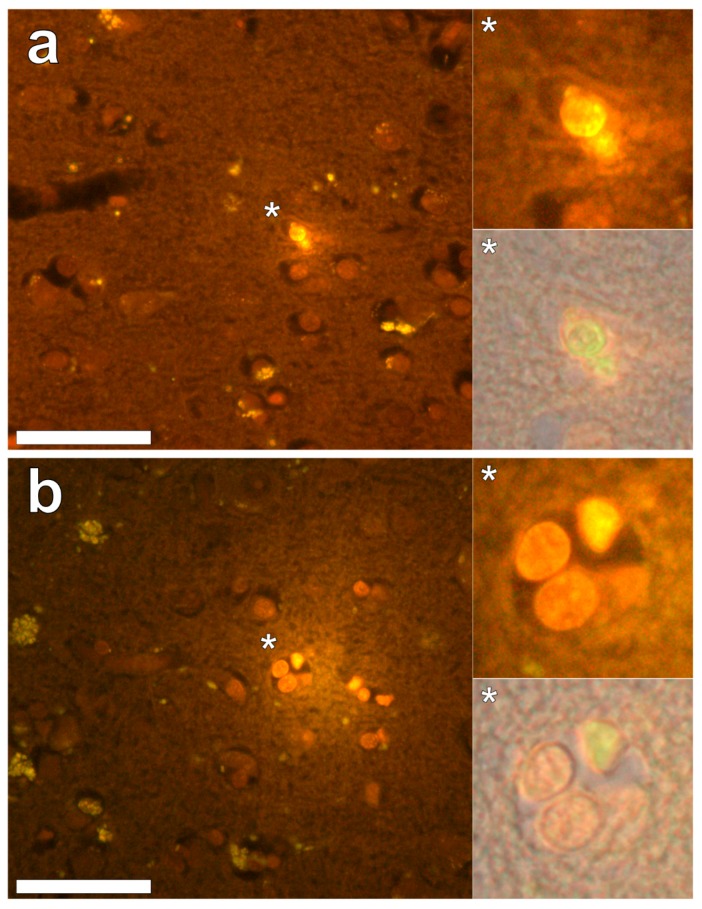
Intracellular aluminium in microglia in close proximity to extracellular deposits of aluminium-rich debris (**a**) and in astrocyte-like cells (**b**) in the occipital cortex. Magnified inserts are depicted (asterisks) with the bright field overlay shown in lower panels. Magnification × 400, scale bars: 50 μm.

**Figure 3 ijerph-16-02129-f003:**
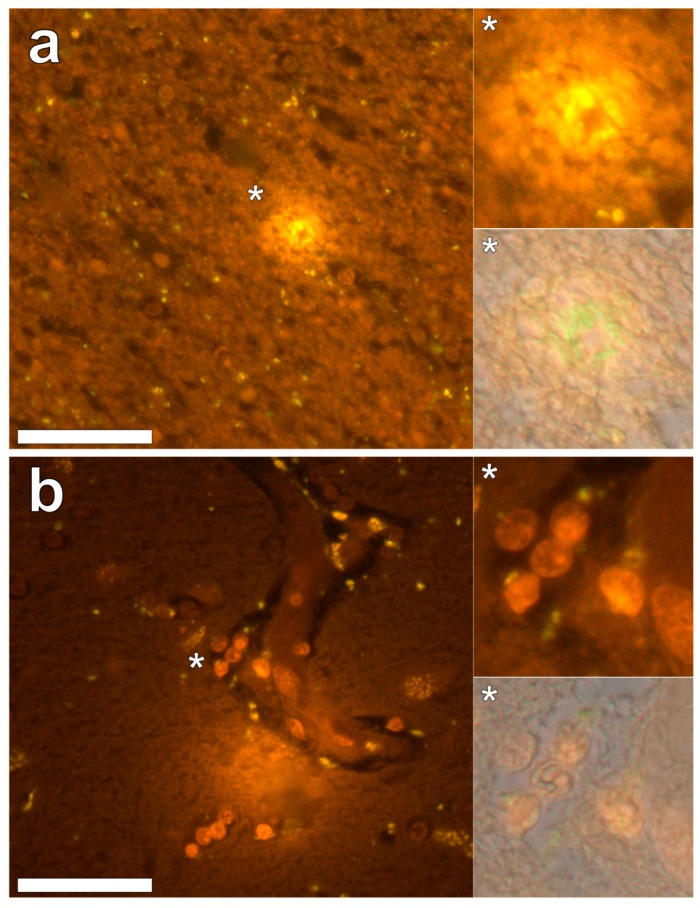
Extracellular aluminium highlighted via an intense orange fluorescence emission in the white matter of the temporal lobe (**a**). Intracellular aluminium was also noted in the temporal cortex (**b**) as highlighted via punctate orange fluorescence within inflammatory cells in the vessel wall. Magnified inserts and their bright field overlays are denoted by asterisks. Magnification × 400, scale bars: 50 μm.

**Figure 4 ijerph-16-02129-f004:**
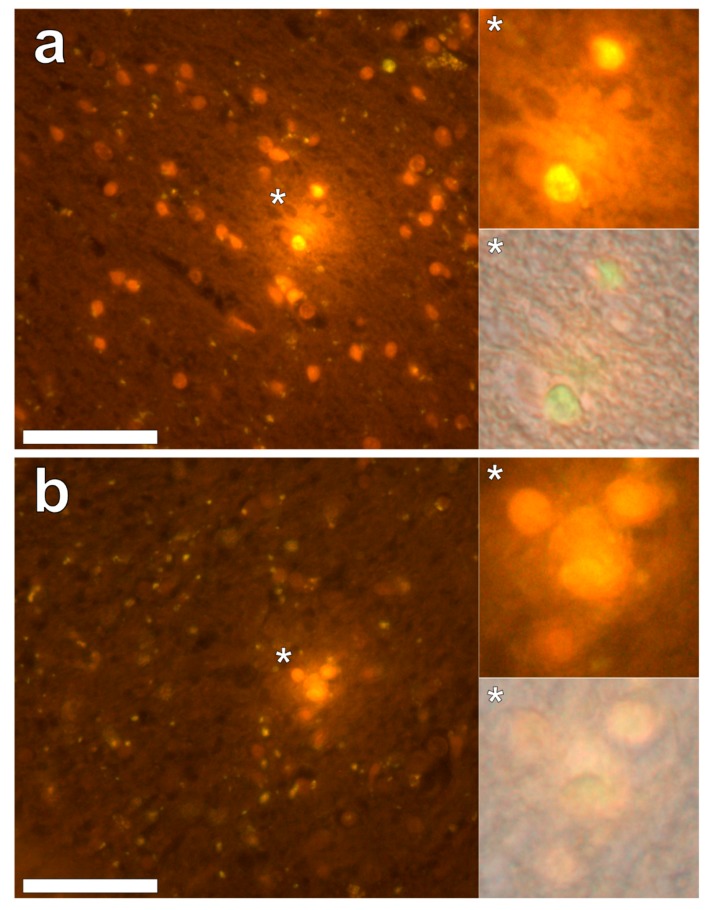
Intracellular aluminium in glial-like cells and within surrounding cellular debris in the parahippocampal gyrus (**a**) and in glial cells exhibiting astrocyte-like processes in the hippocampus (**b**). Magnified inserts are depicted (asterisks) with the bright field channel overlaid. Magnification × 400, scale bars: 50 μm.

**Figure 5 ijerph-16-02129-f005:**
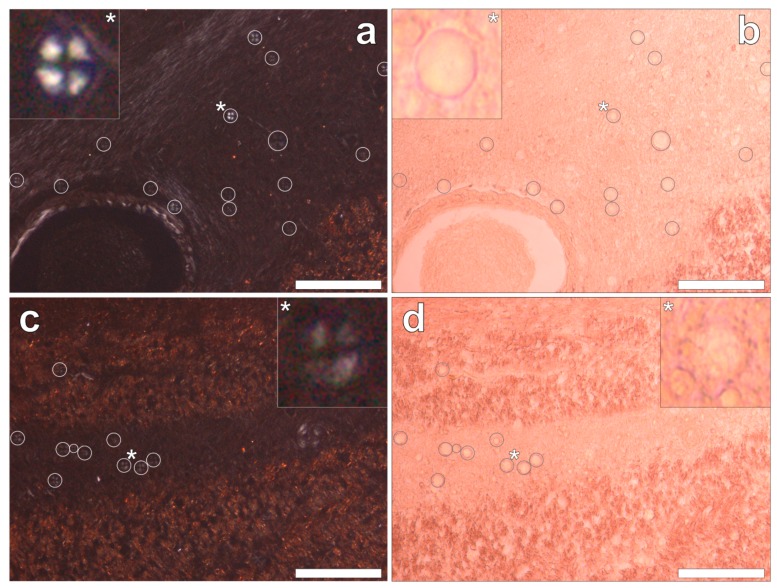
Congo red stained serial tissue sections of the occipital lobe, revealing multiple spherulites. Spherulites were noted via a Maltese-Cross diffraction pattern (white circles) under fully polarised light (**a**,**c**) and an ovoid-like morphology (grey circles) under bright field illumination (**b**,**d**). Spherulites were exclusively distributed across the grey-white matter interface of the occipital lobe. Magnification × 200, scale bars: 100 μm.

**Figure 6 ijerph-16-02129-f006:**
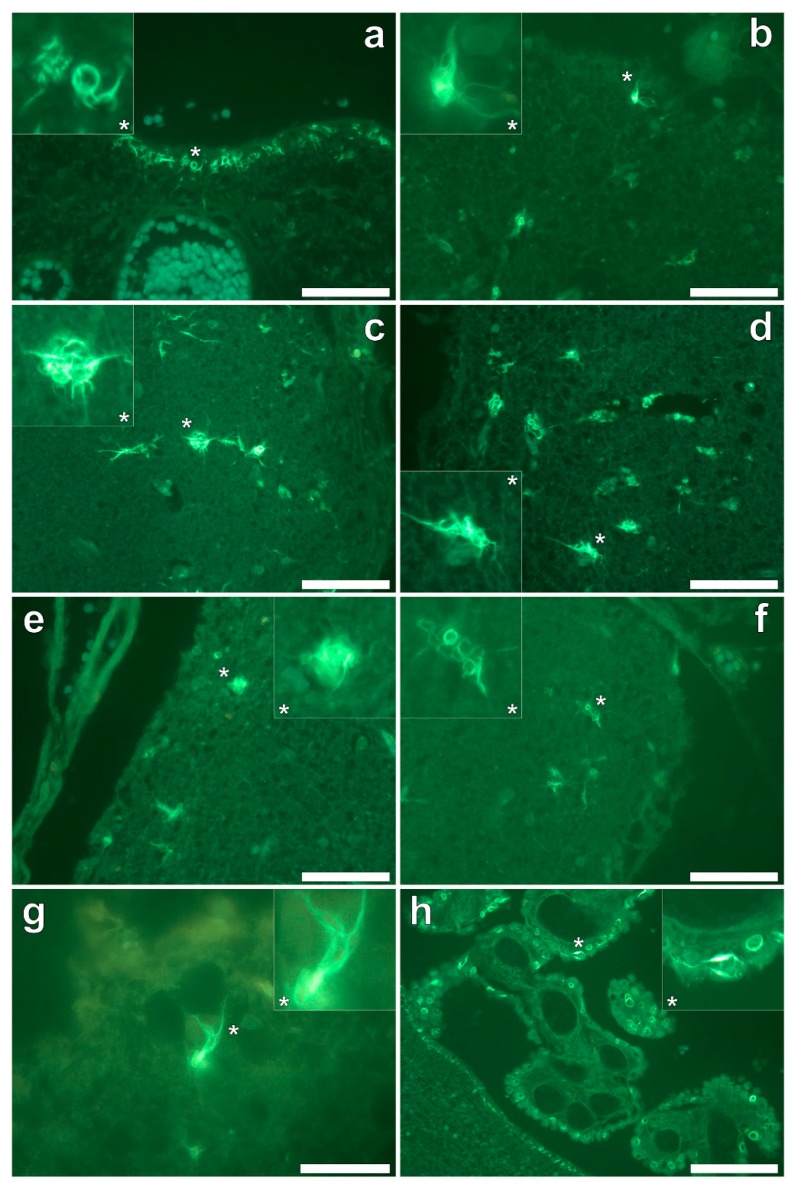
Thioflavin S (ThS) staining of the frontal (**a**,**b**), parietal (**c**,**d**), occipital (**e**,**f**) and temporal lobes (**g**) and hippocampus (**h**) revealed cortical neurofibrillary tangles (NFTs) as highlighted via a green fluorescence emission. ThS-reactive NFTs were noted in pia mater, as demonstrated in the frontal cortex (**a**) and hippocampus (**h**). Ring-like NFTs were also noted in the choroid plexus (**h**). Magnified inserts are denoted with asterisks. Magnifications: (**a**–**f**): × 40, (**g**): × 1000, (**h**): × 200, scale bars: (**a**–**f**): 50 μm, (**g**): 20 μm, (**h**): 100 μm.

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
