# Peer review of "Aluminium in Brain Tissue in Epilepsy: A Case Report from Camelford"

_ijerph, 2019, doi:10.3390/ijerph16122129_

Round 1
Reviewer 1 Report
My comments are:
1)The authors have to clarify the meaning of the number in parenthesis in relation to the Aluminium content either in the Abstract and in the Results section. The number 4.80(2.20) microg/g dry wt represents the mean (SD)?
2)The results related to Aluminium content in brain tissues.can be improved.
Author Response
1)The authors have to clarify the meaning of the number in parenthesis in relation to the
Aluminium content either in the Abstract and in the Results section. The number 4.80(2.20)
microg/g dry wt represents the mean (SD)?
We are grateful to the reviewer for their comments. We have now made it clear that the data
presented in both the Abstract and Results sections in relation to the aluminium contents state
the mean and standard deviation (mean, SD, n=5).
2)The results related to Aluminium content in brain tissues.can be improved.
We would like to thank the reviewer for this suggestion. We have improved the text in the
results related to the aluminium content by minimising repetition in the text.
Reviewer 2 Report
Manuscript ID: ijerph-508342
Type of manuscript: Case Report
Title: Aluminium in brain tissue in epilepsy: A case report from Camelford
This Case Report describes several tests performed on different portions of brain tissue of a 60-year-old man who died of asphyxiation brought on by an epileptic fit. This man was part of the population of the Cornish town of Camelford, which in 1988 was subjected to a very high aluminium exposure through drinking water. With the results obtained, the authors intend to prove the role of aluminium in the death of this man.
With this work the authors demonstrated a correlation between several specific microscopy tests performed in different brain areas and the aluminium concentration obtained by Atomic Absorption in the same zones, especially the hippocampus and occipital lobe
The main conclusion I draw from this work, and which is also mentioned by the authors, is the fact that it is important not to forget this incident and to continue the investigations.
The manuscript is concisely and clearly written, the presentation of data is good and well discussed. Consequently, I recommend to accept this Case Report as it is for publication in the “International Journal of Environmental Research and Public Health” journal.
My observations are minor and given under;
1. The results of the concentration of aluminium obtained by Atomic Absorption would be easier/quicker to observe if they were presented in table instead of text.
2. Keywords: occipital instead of “occiptal”
Author Response
We are grateful for the positive comments of the reviewer.
My observations are minor and given under;
1. The results of the concentration of aluminium obtained by Atomic Absorption would be
easier/quicker to observe if they were presented in table instead of text.
We thank the reviewer for this suggestion. A table was considered, however, as the data
collected is from a single male donor in this case, we believe that the representation of the
data in this work would be better placed in the text. We have removed repetition where
possible for the results relating to the aluminium concentrations in the revised version of the
manuscript.
2. Keywords: occipital instead of “occiptal”
We would like to thank the reviewer for noting this typographical error. We have now
corrected the keyword to “occipital” in the revised version of this manuscript.